# Lattice-Based Group Signature with Message Recovery for Federal Learning

**Yongli Tang** [1], **Deng Pan** [1,*] , **Panke Qin** [1] **and Liping Lv** [2]

1  School of Software, Henan Polytechnic University, Jiaozuo 454000, China
2  Information Engineering Institute, Jiaozuo University, Jiaozuo 454000, China
*  Correspondence: pandeng@home.hpu.edu.cn

**Abstract:** Federal learning and privacy protection are inseparable. The participants in federated learning need to be the targets of privacy protection. On the other hand, federated learning can also be used as a tool for privacy attacks. Group signature is regarded as an effective tool for preserving user privacy. Additionally, message recovery is a useful cryptographic primitive that ensures message recovery during the verification phase. In federated learning, message recovery can reduce the transmission of parameters and help protect parameter privacy. In this paper, we propose a lattice-based group signature with message recovery (GS-MR). We then prove that the GS-MR scheme has full anonymity and traceability under the random oracle model, and we reduce anonymity and traceability to the hardness assumptions of ring learning with errors (RLWE) and ring short integer solution (RSIS), respectively. Furthermore, we conduct some experiments to evaluate the sizes of key and signature, and make a performance comparison between three lattice-based group signature schemes and the GS-MR scheme. The results show that the message–signature size of GS-MR is reduced by an average of 39.17% for less than 2000 members.

**Keywords:** group signature; federal learning; lattice; message recovery; privacy-preserving



## 1. Introduction

Federated learning is a decentralized machine learning paradigm that enables collaborative model training without the need for centralized data aggregation. Multiple parties, such as devices or organizations, participate by computing model updates or gradients locally and exchanging them with a central server [1]. Due to its characteristics, federated learning has gained increasing attention, particularly in the fields of healthcare, finance, and the Internet of Things (IoT) [2–4]. However, in federated learning, the importance of protecting the privacy of participants cannot be overlooked [2]. Therefore, protecting sensitive information becomes a challenging task in the ever-evolving landscape of federated learning. Based on this premise, group signatures have emerged as an effective tool for protecting user privacy due to their anonymity and traceability properties.

Group signature, as a special type of digital signature [5], is a research hotspot in public key cryptography. In the group signature scheme, each member of the group is issued with a signing key, allowing them to generate signatures anonymously by using the signing key (anonymity); if there is an abuse of signature power by malicious group members, the group signature scheme has an entity called the group manager, which can break anonymity by deriving specific signatories from the signature. Due to the characteristics of group signature, it can be applied in federated learning to achieve anonymity preservation and parameter integrity, as well as to prevent dishonest participants from transmitting malicious data, etc.

However, in conventional group signatures, to ensure message integrity, signers must send the message along with the signature to the verifier. This poses a significant problem: during the process of verifying the correctness of the signature, the verifier needs to receive

all the parameters of the message–signature pairs, with the message often taking up a significant portion. To address the above problem, Nyberg et al. [6] introduced the concept of message recovery: It enables the sender to avoid sending the signed message and to send only the signature, and it can recover the message while confirming the validity of the signature. This obviously decreases the quantity of information that needs to be transferred, which saves transmission bandwidth. Moreover, it is more convenient for both the sender and the receiver. The implementation of message recovery is considered as an encoding method [7,8], as it involves adding additional information to the signature to achieve message recoverability. Since then, Islam et al. [9] have proposed a signature scheme for message recovery with specified verifiers based on elliptic curves and bilinear pairs, and the scheme was proven to be secure under the stochastic prediction model. In 2020, Kazmirchuk et al. [10] proposed a provably secure elliptic curve-based digital signature authentication scheme with message recovery. Their scheme uses a hash token function instead of a hash function, allowing for reversed signature and verification procedures and message recovery from the signature r-component. In 2013, Tian et al. [11] first introduced the concept of message recovery to lattice-based cryptography. In 2023, Wu et al. [12] proposed an identity-based proxy signature scheme on the lattice, and it also worked with message recovery. The difference between traditional digital signatures and message recovery signatures is shown in Figure 1.

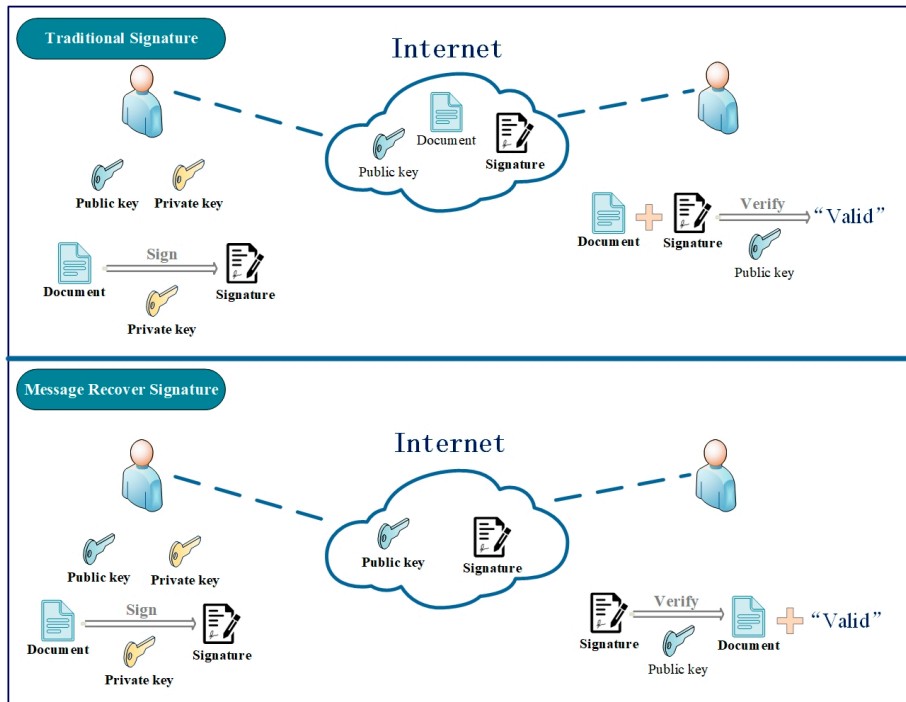

**Figure 1.** The difference between traditional digital signatures and message recovery signatures.

Nevertheless, to the best of our knowledge, the group signature schemes currently proposed do not possess the functionality of message recovery. This will directly result in group members having to send additional messages to the verifier. Therefore, constructing a group signature scheme with message recovery (GS-MR) will reduce the amount of data received by the verifier and provide greater transparency and application scalability to group privacy scenarios. Furthermore, in domains such as federated learning, the GS-MR scheme has significant advantages over traditional group signature schemes. For example, in federated learning, participants train the model locally and send the model parameter updates to a central server, which aggregates these parameter updates and distributes the aggregated model parameters to the participants. The GS-MR scheme can be used to verify the integrity of the model parameters transmitted by the central server and to recover

the original model update information. This ensures that the model parameters are not tampered with during the transmission process and provides verifiability of the results to the participants. In addition, in certain scenarios, such as model analysis or debugging of model updates, federated learning may require the recovery of the original participant data without centralizing the raw data on the central server. The GS-MR scheme can facilitate the recovery of participants' original data from the group signatures, eliminating the need for centralized data collection. This helps protect the privacy of subscriber data and reduces the need for data transmission.

With the continuous breakthroughs in the field of quantum computing, group signature schemes based on traditional number theory constructions are becoming insecure. In 1996, Ajtai [13] introduced the lattice as a cryptographic system with a special algebraic structure. In the post-quantum era, lattice-based cryptography has become a hot research topic in cryptography because of its high asymptotic efficiency, parallelizability, and simplicity of operation. In addition, probabilistic polynomial-time efficient methods for solving difficult problems on lattice do not yet exist under quantum computers [14]. Thanks to the multiple advantages of lattices, Gordon et al. [15] pioneered the construction of the first lattice-based group signature scheme. Gordon's scheme has high theoretical value, but its signature length is too long to be of practical consequence. Ling et al. [16] proposed the first lattice-based constant-size group signature scheme at PKC 2018. They used the "restricted guessing" technique of Ducas and Micciancio's signature scheme [17] and solved the problem of linear growth of the signature size, but the parameters of their scheme were set too large and there were soundness errors in NIZK proof in their scheme. In the subsequent research on lattice-based group signature constructions, numerous improved schemes have been proposed [18–23]. Furthermore, many lattice-based group signature schemes have been proven to be secure in the standard model, such as [24]. However, to the best of our knowledge, a lattice-based GS-MR scheme has not been proposed thus far. Therefore, we aim to construct a lattice-based GS-MR scheme to provide potential security assurance for federated learning scenarios in the quantum era.

*Our Contribution*

We constructed the first lattice-based GS-MR scheme from lattice assumption. In the GS-MR scheme, the message will be recovered in full while the signature is verified as being correct. Therefore, in the rest of this paper, we will use the verification parameter to represent the total size of the message–signature required for the verification phase. The specific contributions are as follows:

(1) We construct a GS-MR based on the Abe-Okamoto signature scheme [25] (ASS) combined with the sign-hybrid-encrypt framework. In the key generation phase, we combine a ring version of Boyen's signature scheme (BSS) with an algorithm for generating ring trapdoors to distribute private signing keys to the group members. In the signature generation phase, the member's identity ID is first encrypted into cipher text using a double encryption algorithm with CCA-security [26] (LPR encryption scheme) to ensure the anonymity and traceability of the group member's identity; The encrypted result is then used as part of the input to the LSS combined with the ASS to generate the final signature.

(2) We prove that the GS-MR scheme satisfies correctness (with message recoverability), full anonymity, and traceability under the random oracle model (ROM). In addition, the anonymity of GS-MR relies on ring learning with errors assumption (RLWE), and the traceability of GS-MR relies on the short integer solution assumption (RSIS).

(3) We have experimentally performed some simple evaluations of the proposed GS-MR scheme, which include a comparison of the key and verification parameters, respectively. Then, we compare three existing lattice-based group signature schemes [19,24,27] with the proposed GS-MR scheme and perform an exhaustive verification parameter size analysis. According to the results of the analysis, the proposed GS-MR scheme reduces the verification parameter size by an average of 39.17%.

The structure of this paper is as follows. In Section 2, we introduce the symbols, lattice, the RSIS and RLWE problems, and some algorithms. In Section 3, we introduce the definition and security model of the GS-MR scheme. Then, we introduce the proposed scheme in Section 4. The security analysis is shown in Section 5. Section 6 presents the efficiency analysis. The last section is a summary of the paper.

## 2. Preliminaries

### 2.1. Symbol Definition

The symbols that appear in this paper are described in Glossary.

### 2.2. Definition of the Lattice

Let $A = \{a_1, a_2, ..., a_m | a_i \in \mathbb{R}^n\}$ be a set of linearly independent column vectors; and the lattice composed of this set of vectors is defined as follows:

$$\Lambda(A) = \mathcal{L}(a_1, a_2, ..., a_m) = \{Ax | x \in \mathbb{Z}^m\}, \tag{1}$$

and $A$ is called the base of lattice $\Lambda(A)$. Most cryptosystems are constructed using an integer lattice, i.e., $a_i \in \mathbb{Z}^n$. If $n = m$, then $\Lambda(A)$ is said to be a full-rank lattice.

**Definition 1.** *Given matrix $A \in \mathbb{Z}_q^{n \times m}$ and $u \in \mathbb{Z}_q^n$, define the following two $q - module$ lattices:*

$$\Lambda_q^{\perp}(A) = \{x \in \mathbb{Z}^m : Ax = 0 \bmod q\}, \tag{2}$$

$$\Lambda_q^u(A) = \{x \in \mathbb{Z}^m | Ax = u \bmod q\}. \tag{3}$$

### 2.3. Ring Variants of the Lattice and the Relevant Difficult Problems

Although lattice-based cryptographic constructions are resistant to quantum attacks, they have not been developed commercially until now because of their low computational efficiency. Due to the use of the expansion of a two-dimensional matrix as an operation, the complexity of the lattice operation is always $O(nm \log q) \approx O(n^2)$. To address this issue, we employed the lattice of ideals, a special algebraic system known as an ideal lattice, and applied SIS and LWE to polynomial ring settings.

**Definition 2.** $RSIS_{n,m,q,\beta}$ *problem. Given $a = (a_1, ..., a_2) \in \mathcal{R}_q^{1 \times m}$, the $RSIS_{n,m,q,\beta}$ is defined as follows: find $x = (x_1, ..., x_m) \in \mathcal{R}_q^m$ satisfying $ax = 0 \bmod q$ and $||x||_\infty \leq \beta$. For $m > \log q / \log(2\beta)$, $\gamma = 16\beta mn \log^2 n$, and $q \geq \gamma \sqrt{n}/4 \log n$, the $RSIS_{n,m,q,\beta}$ problem is as difficult as the $Ideal - SVP_\gamma$ problem [28].*

**Definition 3.** $RLWE_{n,m,q,\chi}$ *problem. Define a vector $s \in \mathcal{R}_q$ and a distribution $\chi$ on $\mathcal{R}$. Given $e \leftarrow \chi$ and a randomly chosen $A \in \mathcal{R}_q$ to obtain $(A, As + e)$, the $RLWE_{n,m,q,\chi}$ is defined as finding an $s \in \mathcal{R}_q$ from $(A, As + e)$. $(A, As + e)$ and $(A, y)$ are indistinguishable, where $A \in \mathcal{R}_q$, and $y \leftarrow \mathcal{R}_q$. The $RLWE_{n,m,q,\chi}$ problem is at least as difficult as the $Ideal - SVP_\gamma$ problem [26].*

### 2.4. Boyen's Signature Algorithm and Its Ring Variants

The BSS [29] is a hybrid algorithm on the lattice. The parameters of BSS are as follows: given security parameter $\lambda$ and message length $\ell$, let $q = ploy(n)$, $m \geq 2n \log q$, $\sigma = \Omega(\sqrt{\ell n \log q})$ and $\beta = \sigma \omega(\sqrt{\log m})$. The BSS key generation algorithm is as follows:

(a) The algorithm $TrapGen(n, m, q)$ [30] produces an $A$ and a trapdoor base $T_A$ of $\Lambda^{\perp}(A)$, where $A$ is statistically close to uniform over $\mathbb{Z}_q^{n \times m}$ and $T_A \in \mathbb{Z}^{m \times m}$ is a short basis for $\Lambda^{\perp}(A) = \{x \in \mathbb{Z}^m : A \cdot x = 0 \bmod q\}$.

(b) Randomly chooses matrices $A_0, A_1 .... A_\ell \in \mathbb{Z}_q^{n \times m}$ and vector $u \in \mathbb{Z}_q^n$.

(c)　Output the public key $PK = (A, A_0, A_1 \ldots . A_\ell, u)$ and the signing key $sk = T_A$.

The BSS signature algorithm is as follows: Upon the input of a fixed-length message $d = (d_1, \ldots . d_\ell) \in \{0, 1\}^\ell$, the signature algorithm first computes $A_{(d)} = [A|A_0 + \sum_{i=1}^\ell d_i A_i] \in \mathbb{Z}_q^{n \times 2m}$; then, it runs the lattice base delegation algorithm $ExtBasis(A_{(d)}, T_A)$ [31] to generate a short base $T_{(d)}$ of $\Lambda^\perp(A_{(d)})$, and finally runs the preimage sample algorithm $Sample(T_{(d)}, A_{(d)}, u, \sigma)$ [30] to obtain a signature $z \in \mathbb{Z}^{2m}$, satisfying $||z||_\infty \leq \beta$ and $A_{(d)}z = u \bmod q$.

By applying BSS to the polynomial ring setting [32] and setting the parameter $m$ to $m = \Omega(\log q)$, the signature size can be reduced from $\ell O(n^2)$ to $\ell O(n)$. The signing public key in the ring variant of BSS is $PK = (a, a_0, \ldots . a_\ell, u) \in (\mathcal{R}_q^m)^{\ell+2} \times \mathcal{R}_q$ and the signing key is $sk = T_a \in \mathbb{Z}^{nm \times nm}$. The security of the ring variant of BSS is based on the difficulty of $RSIS_{n,m,q,\beta}$, which can be reduced to the hardness assumptions of $SVP_{\ell \cdot \tilde{O}(n^2)}$.

*2.5. Gaussian Distribution and Rejection Sampling*

**Definition 4.** *Given any $\sigma > 0$ and vector $c \in \mathbb{R}^m$, the Gaussian distribution centered on $c$ is defined as follows: $D_{\sigma,c}^m = \exp(-\pi||x - c||^2/\sigma^2)/\sum_{x \in \mathbb{Z}} \rho_{\sigma,c}^m(x)$. Gaussian distributions on $\mathbb{R}^m$ are abbreviated as $D_\sigma^m$ when $c = 0$. In the GS-MR scheme, $x \leftarrow D_\sigma^m$ is defined over $\mathcal{R}_q$, which means that every coefficient of $x$ obey distribution $D_\sigma^m$.*

**Lemma 1.** *[33]. Given any $\sigma$ and a positive integer $m$, the following equations are satisfied:*
(1)　$\Pr[x \leftarrow D_\sigma^m : ||x|| > 2\sigma\sqrt{m}] < 2^{-m/4}$.
(2)　$\Pr[x \leftarrow D_\sigma^1 : ||x|| > \sigma k] < 2^{-k^2/2}$

**Lemma 2.** *[34]. Rejection sampling algorithm. Let $V = \{v \in \mathbb{Z}^m : ||v|| < t\}$, $\sigma = \omega(t\sqrt{\log m})$, and $h : V \to \mathbb{R}$ and there exists a universal upper bound $M \in \mathbb{R}$. Then, the statistical distance between the output distributions of the following two algorithms is less than $2^{-\omega(\log m)}/M$.*

*2.6. Key Generation-Related Algorithms*

**Lemma 3.** *[16]. Trapdoor generation algorithm $TrapGen_{\mathcal{R}_q}(n, m, q)$. On input parameters $n$, $m$ and a prime $q$, the algorithm outputs a polynomial vector $a \in \mathcal{R}_q^{1 \times m}$, and a set of parametrically smaller bases $T_a \in \mathbb{Z}^{nm \times nm}$ on the lattice $\Lambda_q^\perp(Rot(a))$, where $Rot(a)$ and $\mathbb{Z}^{n \times nm}$ are statistically close in distribution and satisfy $||T_a|| \leq O(\sqrt{n \log q})$.*

**Lemma 4.** *[35]. Lattice base delegation algorithm $BasisDel(A, R, T_A, \sigma)$. On input $A \in \mathbb{Z}^{n \times m}$, a base $T_A$ of $\Lambda^\perp(A)$, an invertible matrix $R \in \mathbb{Z}^{m \times m}$, and a standard deviation $\sigma \geq ||\tilde{T}_A|| \cdot (\sigma_R\sqrt{m} \cdot \omega(\log^{3/2} m))$, where $\sigma_R = \sqrt{n \log q}\omega(\sqrt{\log m})$, the algorithm outputs a base $T_B$ of $\Lambda^\perp(B)$, where $B = AR^{-1}$, and $||\tilde{T}_B|| < \sigma/\omega(\sqrt{\log m})$.*

**Lemma 5.** *[16]. Preimage sample algorithm $SamplePre_{\mathcal{R}_q}(a, T_a, u, \sigma)$. On input $a \in \mathcal{R}_q^{1 \times m}$ and a base $T_a$ of $\Lambda^\perp(Rot(a))$, a Gaussian parameter $\sigma$, and any polynomial vector $u \in \mathcal{R}$, there exists a algorithm $SamplePre_{\mathcal{R}_q}(a, T_a, u, \sigma)$, which outputs a polynomial vector $e \in \mathcal{R}^m$ satisfying $ae = u \bmod q$.*

### 3. Definition of GS-MR Scheme and Security Model

*3.1. Definition*

A GS-MR scheme contains four probabilistic polynomial time (PPT) algorithms:

(1) $KeyGen(1^\lambda, 1^N)$ : this takes the security parameter $\lambda$ and maximum group members $N$ as the inputs, and outputs the group public key $gpk$, group member's signing key $gsk$, and group manager's tracking key $gtk$.

(2) $Sign(gpk, M, gsk_\pi, \{ID_i\}_{i=1}^N)$ : this takes the group public key $gpk$, a message $M$, the signing key $gsk_\pi$, and a group member's identity set $\{ID_i\}_{i=1}^N$ as the inputs, and outputs a signature $SIG$ of $M$ under $gsk_\pi$.

(3) $Verify(gpk, SIG, \{ID_i\}_{i=1}^N)$ : this takes the public key $gpk$, a signature $SIG$, and a group member's identity set $\{ID_i\}_{i=1}^N$ as the inputs, and outputs "Valid" and complete message $M$ if the signature $SIG$ is a valid signature on the message $M$, or "Invalid" otherwise.

(4) $Open(gpk, SIG, gtk)$ : this takes the group public key $gpk$, a signature $SIG$, and the tracking key $gtk$ as the inputs, outputs the member identity $ID_\pi$ of the signer if the signature $SIG$ is "Valid", checked using $Verify$, or $\perp$ otherwise.

*3.2. Security Model*

In the GS-MR scheme, three properties are required: correctness, anonymity, and traceability. Correctness includes validation correctness, open correctness, and message recoverability, where validation correctness means that the group signature output by the signature algorithm can be successfully verified, recoverability means that the complete signed message can be recovered when the group signature is successfully verified, and open correctness denotes the ability to acquire the right signer's identity from a valid signature. We will describe the strong anonymity of the GS-MR scheme through the CCA (Chosen Ciphertext Attack) security model, as detailed in Definition 5. We will describe the traceability of the GS-MR scheme using Definition 6. To describe the security model of GS-MR, the present paper leverages the security definitions for group signatures of varying strengths provided by Bellare et al. [36]. Through a corresponding game between a challenger $\mathcal{S}$ and an adversary $\mathcal{A}$, the anonymity and traceability guaranteed by the GS-MR scheme will be depicted.

We summarize three distinct query types that an adversary $\mathcal{A}$ can ask in the corresponding games, as well as the possible responses that the challenger $\mathcal{S}$ can give to those queries.

(a) Corrupt query: $\mathcal{A}$ makes a corrupt query on a member's index $i \in [N]$ and $\mathcal{S}$ returns a corresponding signing key $gsk_i$.

(b) Signing query: $\mathcal{A}$ makes a signing query on an index $i$ and a message $M$, and $\mathcal{S}$ runs the algorithm $Sign(gpk, M, gsk_i) \rightarrow SIG$, and returns the signature $SIG$ to $\mathcal{A}$.

(c) Opening query: $\mathcal{A}$ makes an opening query on a signature $SIG$, and $\mathcal{S}$ calls the algorithm $Open(gpk, SIG, gtk)$ to output a member identity $ID_i$, and returns the member's identity $ID_i$ to $\mathcal{A}$; otherwise, it returns to $\perp$.

**Definition 5.** *(Full Anonymity) The property of anonymity in the GS-MR scheme implies that signatures produced by any two distinct signers are computationally indistinguishable. The GS-MR scheme meets full anonymity if for any PPT adversary $\mathcal{A}$, $\mathcal{A}$'s advantage in GAME I in Figure 2 can be negligible.*

GAME I (Anonymity Game):
- KeyGen. Given $\lambda$ and $N$, the challenger $\mathcal{S}$ runs $KeyGen(1^\lambda, 1^N)$ and returns $gpk, gsk$ to $\mathcal{A}$.
- Query. $\mathcal{A}$ adaptively conducts signing and opening queries.
- Challenge. $\mathcal{A}$ submits two identities $i_0, i_1 \in [N]$ and message $M$. $\mathcal{S}$ chooses randomly $b \in \{0,1\}$ and returns $Sign(gpk, M, gsk_{i_b}) \rightarrow SIG$ to $\mathcal{A}$.
- Guess. $\mathcal{A}$ gives $b' \leftarrow \{0,1\}$. $\mathcal{A}$ wins GAME I, if $b' = b$ and never conducts opening query on $M$ and $SIG$.

GAME II (Traceability Game):
- KeyGen. Same as GAME I, except that $\mathcal{S}$ returns $gpk, gtk$ to $\mathcal{A}$.
- Query. $\mathcal{A}$ adaptively conducts corrupt and signing queries.
- Forge. $\mathcal{A}$ outputs a signature $SIG^*$. $\mathcal{A}$ wins GAME II, if $Verify(gpk, SIG^*) = valid$, and any one of the following conditions is satisfied:
  1) $Open(gtk, SIG^*) = \perp$,
  2) $Open(gtk, SIG^*) = i \notin \mathcal{C}$ and $\mathcal{A}$ never conducts signing query for $i$.

**Figure 2.** The definitions for the anonymity game and traceability game.

**Definition 6.** *(Traceability) The property of traceability in the GS-MR scheme implies that the advantage of generating a non-openable signature or blaming for other members is negligible. The GS-MR scheme meets traceability if for any PPT adversary $\mathcal{A}$, $\mathcal{A}$'s advantage in GAME II in Figure 2 can be negligible.*

## 4. Scheme Construction

In the proposed GS-MR scheme, each group member has a fixed length of identity information $\boldsymbol{ID} = (d_1, d_2, ..., d_l) \in \{0,1\}^l$. The parameters of GS-MR are as follows: Let $\lambda$ be the security parameter, and $N$ be the maximum group members. Specifically, let Gaussian parameters $\sigma_1 = poly(n)$ and $\sigma_2 = nm\omega(\log^2 m)\log q$ and modulus $q \geq \beta\omega(n\log n)$ be prime, where $\beta = ploy(n)$ and $m > \log q / \log(2\sigma_1\sqrt{2n})$. The noise boundary of $RLWE_{n,m,q,\chi}$ is set to an integer $b$ and satisfies $b = \widetilde{O}(n^{5/4})$ and $q/b = \ell\widetilde{O}(n)$. Choose four hash functions: $H_1 : \mathbb{Z}_q^n \rightarrow \{0,1\}^{l_1+l_2}, H_2 : \{0,1\}^* \rightarrow \{\boldsymbol{v} \in \{-1,0,1\}^m, ||\boldsymbol{v}|| \leq t\}$, $F_1 : \{0,1\}^{l_2} \rightarrow \{0,1\}^{l_1}$ and $F_2 : \{0,1\}^{l_1} \rightarrow \{0,1\}^{l_2}$ to be modeled as random oracles.

The GS-MR scheme we proposed is as follows:

$KeyGen(1^\lambda, 1^N)$: given $\lambda$ and $N$, the group manager performs Algorithm 1.

---

**Algorithm 1:** $KeyGen(1^n, 1^N)$

---

1: $(\boldsymbol{a}, \boldsymbol{T_a}) \leftarrow TrapGen_{\mathcal{R}_q}(n, m, q)$
2: Randomly choose $\boldsymbol{a_0}, \boldsymbol{a_1} \dots \boldsymbol{a_l} \leftarrow \mathcal{R}_q^{m \times m}$
3: **for all** $1 \leq i \leq N$ **do**
4:     $\boldsymbol{D_i} = \boldsymbol{a} \cdot \left(\sum_{j=1}^l d_j \boldsymbol{a_j}\right)^{-1}$
5:     $\boldsymbol{T_{D_i}} \leftarrow BasisDel(Rot(\boldsymbol{a}), \sum_{j=1}^l Rot(\boldsymbol{a_j})d_j, \boldsymbol{T_a}, \sigma_2)$
6:     $\boldsymbol{e_i} \leftarrow SamplePre_{\mathcal{R}_q}(\boldsymbol{D_i}, \boldsymbol{T_{D_i}}, u, \sigma_2)$, such that $\boldsymbol{D_i}\boldsymbol{e_i} = u \bmod q$
7:     $gsk_i = \boldsymbol{e_i}$
8: **end for**
9: Randomly choose $u \leftarrow \mathcal{R}$, $f \leftarrow \mathcal{R}_q$, $s \leftarrow \chi$, $e \leftarrow \chi$
10: Calculate $g = (f \otimes s + e) \bmod q$
11: **Output** : $gpk = [\boldsymbol{a}, \boldsymbol{a_0}, \boldsymbol{a_1} ... \boldsymbol{a_l}, u, f, g]$, $gtk = s$, $gsk = \{\boldsymbol{e_i}\}_{i=1}^N$

---

$Sign(gpk, M, gsk_\pi, \{\boldsymbol{ID_i}\}_{i=1}^N)$: given $gpk$, message $M \in \{0,1\}^{l_2}$, signing key $gsk_i = \boldsymbol{e_i}$ and a group member's identity set $\{\boldsymbol{ID_i}\}_{i=1}^N$, the signer runs Algorithm 2.

---

**Algorithm 2:** $Sign(gpk, M, gsk_\pi, \{\boldsymbol{ID}_i\}_{i=1}^N)$

---

1: **for all** $1 \le i \le N$ **do**
2:       $\boldsymbol{y}_i \leftarrow D_{\sigma_1}^m$
3:       Calculate $\boldsymbol{D}_i = \boldsymbol{a} \cdot (\sum_{j=1}^l d_j \boldsymbol{a}_j)^{-1}$
4: **end for**
5: Calculate $\alpha = H(\sum_{j=1}^N \boldsymbol{D}_j \boldsymbol{y}_j)$
6: Calculate $M' = F_1(M) || (F_2(F_1(M)) \oplus M)$
7: Calculate $r = M' \oplus \alpha$
8: Expend $\boldsymbol{ID}_\pi$ **to** $\overline{\boldsymbol{ID}}_\pi = (0^{n-l} || \boldsymbol{ID}) \in \{0, 1\}^n$
9: $ID'_\pi = \tau^{-1}(\overline{\boldsymbol{ID}}_\pi) \in \mathcal{R}$
10:  Randomly choose $w \leftarrow \chi, e_1, e_2 \leftarrow \chi$
11: Calculate $(c_1 = f \otimes w + e_1, c_2 = g \otimes w + e_2 + [q/2]ID'_i) \in \mathcal{R}_q^2$
12: $\boldsymbol{v} = H_2(r, c_1, c_2)$
13: **for all** $1 \le i \le N$ **do**
14:       **if** $i \ne \pi$, **then** $z_i = \boldsymbol{y}_i$
15:       **else calculate** $z_i = e_i \boldsymbol{v} + \boldsymbol{y}_i$, and outputs $(z_i, r)$ with probability
$\min(1, D_{\sigma_1}^m(z_i)/MD_{e_i v, \sigma_1}^m(z_i))$.
16: **end for**
17: $\Pi = \{z_i\}_{i=1}^N$
18: **Output:** $SIG = (\Pi, (c_1, c_2), r)$

---

$Verify(gpk, SIG, \{\boldsymbol{ID}_i\}_{i=1}^N)$ : given $gpk$, signature $SIG$ and identity set $\{\boldsymbol{ID}_i\}_{i=1}^N$, the verifier performs Algorithm 3.

---

**Algorithm 3:** $Verify(gpk, SIG, \{\boldsymbol{ID}_i\}_{i=1}^N)$

---

1: Parse $\Pi = (z_1, z_2, \ldots, z_N)$
2: **for all** $1 \le i \le N$ **do**
3:       Calculate $\boldsymbol{D}_i = \boldsymbol{a} \cdot (\sum_{j=1}^l d_j \boldsymbol{a}_j)^{-1}$
4: **end for**
5: Calculate $\alpha = H_1(\sum_{i=1}^N \boldsymbol{D}_i z_i - u H_2(r, c_1, c_2))$
6: Set $M' = \alpha \oplus r$
7: Set $M = |M'|_{l2} \oplus F_2(|M'|^{l1})$
8: **if** $F_1(M) = |M'|^{l_1}$ and $||z|| \le 2\sigma\sqrt{m}$ **then**
9:       **return** "Valid"
10: **else**
11:        **return** "Invalid"
12: **end if**

---

$Open(gpk, SIG, gtk)$ : given $gpk$, signature $SIG$, and tracking key $gtk$, the group manager performs Algorithm 4.

---

**Algorithm 4:** $Open(gpk, SIG, gtk)$

---

1: $ID^* = (d_i^*) = c_2 - c_1 s$ where $i = 1, \ldots, n$
2: **for all** $i$ such that $1 \le i \le n$ **do**
3:       **if** $d_i^* \approx \lfloor q/2 \rfloor$ **then**
4:             $d'_i = 1$
5:       **else**
6:             $d'_i = 0$
7: **end for**
8: $ID' = (d'_1, d'_2, \ldots, d'_n)$
9: $\overline{\boldsymbol{ID}}_\pi = \tau(ID') = (\overline{d}_0, \overline{d}_1, \ldots, \overline{d}_{n-1})^T \in \mathbb{Z}_q^n$
10: **if** $\overline{\boldsymbol{ID}}_\pi$ satisfy format $(0^{n-l} || \boldsymbol{ID})$ **then**
11:       **return** $\boldsymbol{ID}$
12: **else**
13:       **return** $\perp$

---

## 5. Security Analysis

We prove that the proposed GS-MR scheme satisfies correctness (validation correctness, message recoverability, and open correctness), full anonymity, and traceability.

**Theorem 1 (correctness).** *The GS-MR scheme is correct.*

**Proof of Theorem 1.**

(1)    Verification correctness and message recoverability.

Given a legal and valid signature, the following equation holds:

$$
\begin{aligned}
\sum_{i=1}^{N} \boldsymbol{D}_i \boldsymbol{z}_i - u H_2(r, c_1, c_2) &= \sum_{i \in [N] \setminus \{\pi\}} \boldsymbol{D}_i \boldsymbol{y}_i + \boldsymbol{D}_\pi \boldsymbol{z}_\pi - u H_2(r, c_1, c_2), \\
&= \sum_{i \in [N] \setminus \{\pi\}} \boldsymbol{D}_i \boldsymbol{y}_i + \boldsymbol{D}_\pi (\boldsymbol{e}_\pi \boldsymbol{v} + \boldsymbol{y}_\pi) - u H_2(r, c_1, c_2), \\
&= \sum_{i=1}^{N} \boldsymbol{D}_i \boldsymbol{y}_i + u \boldsymbol{v} - u H_2(r, c_1, c_2), \\
&= \sum_{i=1}^{N} \boldsymbol{D}_i \boldsymbol{y}_i.
\end{aligned}
\tag{4}
$$

Then, we have

$$
M' = H_1\left(\sum_{i=1}^{N} \boldsymbol{D}_i \boldsymbol{y}_i\right) \oplus r = H_1\left(\sum_{i=1}^{N} \boldsymbol{D}_i \boldsymbol{z}_i - u H_2(r, c_1, c_2)\right) \oplus r,
\tag{5}
$$

and since

$$
M' = F_1(M) || (F_2(F_1(M)) \oplus M),
\tag{6}
$$

we can recover the message

$$
M = [M']_{l_2} \oplus F_2([M']^{l_1})
\tag{7}
$$

From it, and the message $M$ must satisfy $F_1(M) = [M']^{l_1}$. On the other hand, when $i \neq \pi$ and $\boldsymbol{z}_i = \boldsymbol{y}_i$, where $\boldsymbol{y}_i \leftarrow D_{\sigma_1}^m$, according to Lemma 1, $||\boldsymbol{z}_i|| \in [N]_{/\{\pi\}}$ satisfies $||\boldsymbol{z}_i|| \leq 2\sigma\sqrt{m}$ with overwhelming probability; when $i = \pi$, we have $\boldsymbol{z}_i = \boldsymbol{e}_i \boldsymbol{v} + \boldsymbol{y}_i$, and where $\boldsymbol{y}_i \leftarrow D_{\sigma_1}^m$, according to Lemma 2, $\boldsymbol{z}_i$ is statistically indistinguishable from Gaussian distribution $D_{\sigma_1}^m$. Therefore, for all $\boldsymbol{z}_i$, $||\boldsymbol{z}_i|| \leq 2\sigma\sqrt{m}$ is established with overwhelming probability.

(2)    Opening correctness

The correctness of opening depends on the accuracy of the underlying LPR encryption, and the parameter settings described in Section 4. of this paper meet the correctness requirements of the encryption scheme, assuming that $SIG = (\Pi, (c_1, c_2), r)$ is a signature generated by an honest member $i$ through the algorithm $Sign(gpk, M, gsk_i, \{\boldsymbol{ID}_i\}_{i=1}^{N})$. Regarding the validity of the open algorithm, we show that

$$
\begin{aligned}
c_2 - c_1 s &= g \otimes w + e_2 + \lfloor q/2 \rfloor ID'_i - (f \otimes w + e_1) \otimes s \\
&= (f \otimes s + e) \otimes w + e_2 + \lfloor q/2 \rfloor ID'_i - (f \otimes w + e_1) \otimes s \\
&= ew + e_2 - e_1 s + \lfloor q/2 \rfloor ID'_i,
\end{aligned}
\tag{8}
$$

where $||w||_\infty, ||e_1||_\infty, ||e_2||_\infty \leq b$. Note that $b = \widetilde{O}(n^{5/4})$ and $q/b = \ell\widetilde{O}(n)$, it can therefore be concluded that $||ew + e_2 - e_1 s||_\infty \leq 2n \cdot b^2 + b = \widetilde{O}(n^{3.5}) \ll \lfloor q/10 \rfloor$. Next, we determine the value of the $\boldsymbol{ID}$ based on the value of each component of $c_2 - c_1$. The algorithm $Open(gpk, SIG, gtk)$ then recovereds the $\boldsymbol{ID}$ with a probability of 1. □

**Theorem 2 (full anonymity).** *The GS-MR scheme meets full anonymity under ROM if the $RLWE_{n,m,q,\chi}$ problem is hard.*

**Proof of Theorem 2.** Let $\mathcal{A}$ be any PPT adversary in Definition 1; the following proves that the GS-MR scheme satisfies the anonymity requirement by showing that the four games $G_0$, $G_0'$, $G_1$, $G_1'$ are indistinguishable.

$G_0$ : $G_0$ is denoted as the experiment with $b = 0$ in GAME I. First, the system is set up, and the challenger $\mathcal{S}$ calls the algorithm $KeyGen(1^n, 1^N)$ to generate $gpk$, $gtk$, and $gsk = \{e_i\}_{i=1}^N$. Then, $\mathcal{S}$ sends $gpk$ and $gsk = \{e_i\}_{i=1}^N$ to $\mathcal{A}$. $\mathcal{A}$ is permitted to make the following adaptive queries:

(a) Signing query: $\mathcal{A}$ makes a signing query on an index $i$ and a message $M$, and then $\mathcal{S}$ returns the signature $SIG = (\Pi, (c_1, c_2), r)$ to $\mathcal{A}$.

(b) Opening query: $\mathcal{A}$ makes an opening query on a signature $SIG$, $\mathcal{S}$ calls the algorithm $Open(gpk, SIG, gtk)$ to output a member identity $\textbf{\textit{ID}}_i$, and returns the member's identity $\textbf{\textit{ID}}_i$ to $\mathcal{A}$; otherwise, it returns to $\perp$.

$\mathcal{A}$ selects two indexes $i_0, i_1 \in [N]$ with $i_0 \neq i_1$ and a message $M$, and sends them to $\mathcal{S}$. Then, $\mathcal{S}$ calls the signature algorithm $Sign(PK, M, gsk_{i_0}, \{ID_i\}_{i=1}^N)$ and sends the signature $SIG = (\Pi, (c_1, c_2), r)$ to $\mathcal{A}$.

$G_0'$ : $G_0'$ was modified from $G_0$ When calling the algorithm $Sign(gpk, M, gsk_{i_0}, \{\textbf{\textit{ID}}_i\}_{i=1}^N)$ to generate a signature, calculate $z_{i_0} = e_{i_0} v + y_{i_0}$ and $z_{i_1} = e_{i_1} v + y_{i_1}$ in addition to the remaining steps in the signature generation process (calculate the values of $j \notin \{i_0, i_1\}$ and $z_j$ according to the scheme described above).

$G_1$ : $G_1$ is identical to $G_0$ with the exception that $\mathcal{S}$ chooses $b = 1$ rather than $b = 0$.

$G_1'$ : $G_1'$ is identical to $G_0'$. $\square$

**Lemma 6.** *$G_0$ and $G_0'$ are computationally indistinguishable.*

**Proof of Lemma 6.** In the $G_0$ and $G_0'$ games, the difference in signatures lies in the calculation of $z_{i_1}$. According to Lemma 2, the value $z_{i_1}$ produced via rejection sampling in $G_0'$ is statistically equivalent to the value produced by the Gaussian distribution $D_{\sigma_1}^m$ (statistical distance less than $2^{-\omega(\log m)}/M$). In $G_0$, the value of $z_{i_1}$ is taken from the Gaussian distribution $D_{\sigma_1}^m$ and so the games $G_0$ and $G_0'$ are computationally indistinguishable. $\square$

**Lemma 7.** *$G_1$ and $G_1'$ are computationally indistinguishable.*

**Proof of Lemma 7.** The indistinguishability of computation between $G_1$ and $G_1'$ (where $b = 1$ in the context of the GAME I model) is proved in the same way as described above. $\square$

**Lemma 8.** *If the $RLWE_{n,m,q,\beta}$ problem is hard, $G_0'$ and $G_1'$ are computationally indistinguishable.*

**Proof of Lemma 8.** Games $G_1'$ and $G_0'$ are similar, except for the computation of the part containing member information $c_2$ in the signature (which includes the identity of the member $i_1$). Therefore, it is only necessary to prove that the signature $SIG' = (\Pi', (c_1', c_2'), r')$ generated in $G_0'$ and the signature $SIG* = (\Pi*, (c_1^*, c_2^*), r*)$ in $G_1'$ are indistinguishable in computation.

The $c_2'$ and $c_2^*$ can be seen as the LPR encryption of different member identities ($i_0$ and $i_1$) according to the LPR encryption scheme [26], which is indistinguishable (satisfies IND-CCA security) under the $RLWE_{n,m,q,\beta}$ assumption. Therefore, for an adversary $\mathcal{A}$, $c_2'$ and $c_2^*$ are indistinguishable in computation. Combined with the above proof, $SIG' = (\Pi', (c_1', c_2'), r')$ and $SIG* = (\Pi*, (c_1^*, c_2^*), r*)$ are statistically indistinguishable. Therefore, $G_0'$ and $G_1'$ are indistinguishable in computation. $\square$

In conclusion, $G_0$, $G_0'$, $G_1$, $G_1'$ are statistically indistinguishable. Therefore, for any adversary $\mathcal{A}$, when facing GAME I defined at the beginning of this paper, the advantage of winning the game is $Adv = |\Pr[b' = b] - 1/2| + negl(n)$, indicating that $\mathcal{A}$ has no advantage in winning the anonymity game. It can be inferred that the GS-MR scheme satisfies the anonymity requirement.

**Theorem 3 (full traceability).** *The GS-MR scheme meets full traceability under ROM if the* $RSIS_{n,m,q,\beta}$ *problem is hard.*

**Proof of Theorem 3.** When proving the traceability of the GS-MR scheme, there are two key components: (1) The algorithm *Sign* generates legal signatures that can be traced back to the identities of their signers. (2) No adversary can forge a legal and untraceable group signature. First, as shown in Theorem 1, the GS-MR scheme is proven to correctly open any valid signature and query the identity details of the signer. Therefore, the following proof focuses on the second point, namely, that it is impossible for any PPT adversary to successfully construct a legally and untraceable signature. □

Let $\mathcal{A}$ be any PPT algorithm defined in Definition 6 that can forge a signature with a non-negligible advantage after numerous inquiries. Then, a challenger $\mathcal{S}$ can be built to solve the $RSIS_{n,m,q,\beta}$ problem with a non-negligible advantage.

Let the challenger $\mathcal{S}$ maintain three lists $l_1, l_2, C, \Gamma$, and initialize them as empty. Then, $\mathcal{S}$ honestly runs the algorithm $KeyGen(1^\lambda, 1^N)$ of the scheme, with input security parameter $\lambda$ and maximum member group $\{\boldsymbol{ID}_i\}_{i=1}^N$, randomly selects $j \in \{1, 2, \ldots, N\}$, generates $gpk$, $gsk_j$, $gtk$, and then sends $gpk$ and $gtk$ to $\mathcal{A}$. In response to $\mathcal{A}$'s inquiry, $\mathcal{S}$ replied as follows ($\mathcal{A}$ had conducted relevant $H_1$ queries and $H_2$ queries prior to performing signing and corrupt queries):

(a) $H_1$ query. $\mathcal{A}$ selects N polynomial vectors $\boldsymbol{y}_j \leftarrow D_{\sigma_1}^m$ to $\mathcal{S}$. $\mathcal{S}$ first checks list $L_1$. If $\mathcal{A}$ has previously submitted the same query, $\mathcal{S}$ directly returns the same query result. Otherwise, $\mathcal{S}$ selects a random vector $\alpha \in \{-1, 0, 1\}^{l_1 + l_2}$ and returns it to $\mathcal{A}$. For this query, $\mathcal{S}$ records $(\{\boldsymbol{y}_i\}_{i=1}^N, \alpha)$ in the list $L_1$.

(b) $H_2$ query. $\mathcal{A}$ selects a message $M$, and $c_1$, $c_2$, and submits them to $\mathcal{S}$. $\mathcal{S}$ first checks list $L_2$. If $\mathcal{A}$ has previously submitted the same query, $\mathcal{S}$ directly returns the same query result. Otherwise, $\mathcal{S}$ selects a random vector $v \in \{-1, 0, 1\}^m$ and returns it to $\mathcal{A}$. For this query, $\mathcal{S}$ records $(M, v, c_1, c_2)$ in the list $L_2$.

(c) Corrupt query. $\mathcal{A}$ inputs $k \in [N]$, if $k = j$, $\mathcal{S}$ terminates the game; if $k \neq j$, $\mathcal{S}$ sends the signing key $gsk_k$ to $\mathcal{A}$. For this query, $\mathcal{S}$ records $(k, \boldsymbol{e}_k)$ in the list $C$.

(d) Signing query. $\mathcal{A}$ inputs $k \in [N]$ and message $M$, if $k = j$, $\mathcal{S}$ will modify $\boldsymbol{z}_i$ in the algorithm *Sign* to $\boldsymbol{z}_i = \boldsymbol{y}_i$, and return the signature $SIG$ to $\mathcal{A}$; if $k \neq j$, $\mathcal{S}$ honestly runs the algorithm *Sign* and returns the signature $SIG$ to $\mathcal{A}$. For this query, $\mathcal{S}$ records $(k, M)$ in the list $\Gamma$.

After a series of queries, $\mathcal{A}$ outputs a forged group signature $SIG^* = (\Pi^*, (c_1^*, c_2^*), r^*)$. If the signature $SIG^*$ satisfies Definition 6, it implies that $\mathcal{A}$ wins GAME II. We analyzed the following two aspects:

(1) Assuming that $SIG^*$ is a valid signature and satisfies $Open(gpk, SIG^*, gtk) = j$. Since the signature is valid, it follows that $v^* = H_2(\sum_{i=1}^N \boldsymbol{D}_i \boldsymbol{z}_i - uv^*, c_1, c_2)$; Furthermore, since the signature $SIG$ forged by $\mathcal{A}$ can satisfy the verification correctness, we have $v^* = H_2(\sum_{i=1}^N \boldsymbol{D}_i \boldsymbol{z}_i, c_1^*, c_2^*)$. As the collision probability of the hash-oracle is negligible, we can see that $c_1 = c_1^*$ and $c_2 = c_2^*$, and therefore we can conclude that

$$\sum_{i=1}^N \boldsymbol{D}_i(\boldsymbol{z}_i - \boldsymbol{z}_i^*) = \mathbf{0} \bmod q, \tag{9}$$

since $\sum_{i=1}^N ||\boldsymbol{z}_i - \boldsymbol{z}_i^*|| \leq b$, it follows that $\sum_{i=1}^N (\boldsymbol{z}_i - \boldsymbol{z}_i^*)$ is a solution to the $RSIS_{n,m,q,\beta}$ problem.

(2) Assuming $Open(gpk, SIG^*, gtk) = \bot$, the forged signature $SIG^* = (\Pi^*, (c_1^*, c_2^*), r^*)$ produced by the adversary $\mathcal{A}$ satisfies the following condition

$$\boldsymbol{ID}_k^* \notin \{\boldsymbol{ID}_i\}_{i=1}^N \text{ i.e., } \boldsymbol{D}_k = \boldsymbol{a} \cdot \left(\sum_{i=1}^l d_{i(k)} \boldsymbol{a}_i\right)^{-1} \neq \boldsymbol{a} \cdot \left(\sum_{i=1}^l d_{i(k)}^* \boldsymbol{a}_i\right)^{-1} = \boldsymbol{D}_k^*. \tag{10}$$

Since the above condition can be verified by the algorithm $Verify(gpk, SIG, \{ID_i\}_{i=1}^N)$, we have

$$\sum_{i=1}^N \boldsymbol{D}_i \boldsymbol{z}_i^* - uv^* = \sum_{i=1}^N \boldsymbol{D}_i \boldsymbol{y}_i^* \ i.e. \ \boldsymbol{D}_k^* \boldsymbol{z}_k^* - uv^* = \boldsymbol{D}_k^* \boldsymbol{y}_k^*. \tag{11}$$

Let

$$\boldsymbol{D}_k^* \boldsymbol{z}_k - uv^* = \boldsymbol{D}_k^*(e_k v^* + \boldsymbol{y}_k^*) - uv^* = \boldsymbol{D}_k^* \boldsymbol{y}_k^*. \tag{12}$$

From Equations (11) and (12), we can obtain

$$\boldsymbol{D}_k^*((e_i^* - e_i)v^*) = \boldsymbol{0} \bmod q. \tag{13}$$

Since $||(e_i^* - e_i)v^*||_\infty \leq 4\sigma_1 t\sqrt{m}$, $(e_i^* - e_i)v^*$ is a solution to the $RSIS_{n,m,q,\beta}$ problem.

Based on the proof of the two above cases, if the adversary $\mathcal{A}$ wins GAME II, then the challenger $\mathcal{S}$ will obtain a solution to the $RSIS_{n,m,q,\beta}$ problem. However, the $RSIS_{n,m,q,\beta}$ problem is to solvedifficult under the parameters provided in this paper, and so $\mathcal{A}$ cannot satisfy the two conditions mentioned above. Therefore, the GS-MR scheme has full traceability.

## 6. Implementation and Efficiency Analysis

As proof of concept, in order to understand the practicality of group signatures with recoverable messages, we simply performed some implementations of the GS-MR scheme. We have shown implementations of the GS-MR, which were experimented with using an AMD Ryzen 5 5600G @ 3.90GHz CPU with 16GB of RAM. The programs were compiled using SageMath and Python 3.8. A selection of some program parameters were first shown. Then, based on these parameters, some corresponding outputs in the GS-MR scheme were experimentally derived. In Table 1, we summarize the theoretical estimates of the key size and signature size of GS-MR, where "$n \cdot S$ denotes $n$ elements in a set $S$".

**Table 1.** Theoretical estimation of key size and message–signature size.

|  | Form | Size |
|---|---|---|
| Public Key | $(a, a_0, ..., a_l, u, f, g)$ | $(nm + lnm^2 + 3n)\log_2 q$ |
| Signing Key | $e_i$ | $nm \cdot D_{\sigma_3}$ |
| Tracking Key | $gtk = s$ | $n\log_2 q$ |
| Message–Signature | $(\Pi, (c_1, c_2), r) + M$ | $Nm \cdot D_{\sigma_1} + 2n\log_2 q + l_2$ |

We followed the parameter settings of Luo et al. [37] and also considered the security of the parameter settings in this paper.

- To keep $(\boldsymbol{a}, \boldsymbol{T_a}) \leftarrow TrapGen_{\mathcal{R}_q}(n, m, q)$ working safely, set $q \geq 2$, $m \geq 1$, and $\overline{m} \geq \log q / \log(2\sigma_1\sqrt{2n})$.
- For the Gaussian parameter in $BasisDel(\boldsymbol{A}, \boldsymbol{R}, \boldsymbol{T_A}, \sigma)$, we chose $\sigma_R = \sqrt{n\log q}\omega(\sqrt{\log m})$, and according to Pino [38], let $\sigma_2 \geq ||\tilde{\boldsymbol{T}}_A|| \cdot (\sigma_{\mathbf{R}}\sqrt{m} \cdot \omega(\log^{3/2} m))$.
- For Gaussian parameters in rejection sampling, we chose $\sigma_1 = \omega(t\sqrt{\log m}) \approx 12 \cdot t \cdot \sqrt{nm}$.
- For the choice of $M$ in rejection sampling, according to Definition 4, if $\sigma = 12||\boldsymbol{c}||$, then the probability of $e^{1+1/288}/M \geq D_\sigma^m(x)/M \cdot D_{c,\sigma(x)}^m$ and $e^{1+1/288}/M \geq 3/M$ is greater than $1 - 2^{-100}$. Then, we can fix $M = 3$.

Specifically, we will use the following specific parameters for our experiments:

$pp_1 : q = 2^{24}, n = 512, m = 3, t = 14, \overline{m} = 1536, M = 3, \sigma_1 \approx 6641.8455, \sigma_2 = 1165.2235$.

$pp_2 : q = 2^{27}, n = 512, m = 4, t = 14, \overline{m} = 2048, M = 3, \sigma_1 \approx 7602.8121, \sigma_2 = 1504.2467$.

$pp_3 : q = 2^{29}, n = 1024, m = 3, t = 14, \overline{m} = 3072, M = 3, \sigma_1 \approx 9311.5051, \sigma_2 = 1822.4527$.

In Table 2, we summarize the real output key values of the GS-MR scheme (i.e., public key, signing key, and tracking key) for three specific parameters compared to their theoretical estimates. For a more intuitive display, we plotted Figure 3 to visualize the variations in key sizes under different parameter settings. Based on Table 2 and Figure 3a, it can be observed that the storage cost of our GS-MR scheme mainly lies in the group public key. In Figure 3b, it can be seen that the signing key and tracking key of the GS-MR scheme are only the size of single-digit KB under the three sets of parameters. Although the signed message is not to be used as an input parameter in the verification phase, we also performed a comparative experiment with the signature and message–signature pairs. Subsequently, we conducted 10 experiments for each of the specific group orders 128, 256, 512, and 1024 to compare the signature size and the total length of message-signature under different scenarios (for Section 4, we fixed the value of $l_2 = 2^{19}$). Based on the experimental results, we evaluated the average signature size, as shown in Table 3 and Figure 4, which indicates that the signature size of GS-MR increases linearly with the number of group members. With a fixed message, the proportion of the message size decreases as the group size increases. However, we believe that our GS-MR scheme is still feasible. Particularly in small group environments with a low channel bandwidth and poor communication quality, the proposed GS-MR scheme can ensure a smaller total parameter transmission and alleviate concerns regarding the impact of channel noise on message transmission.

**Table 2.** Public key, signing key and tracking key sizes (in KB).

| Parameters | $PP_1$ | $PP_2$ | $PP_3$ |
|---|---|---|---|
| Public key (Theo.) | 171 | 335 | 414 |
| Public key (Exp.) | 196 | 367 | 443 |
| Signing key (Theo.) | 3.00 | 4.00 | 6.00 |
| Signing key (Exp.) | 3.13 | 4.17 | 6.29 |
| Tracking key (Theo.) | 1.50 | 1.70 | 1.80 |
| Tracking key (Exp.) | 1.55 | 1.75 | 1.87 |

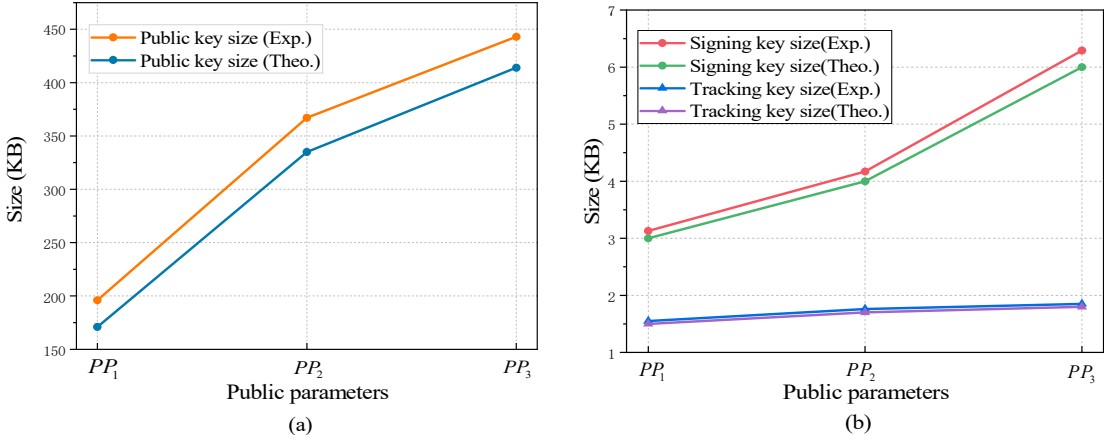

**Figure 3.** Comparison of size in both the theoretical estimation (Theo.) and experiments (Exp.). (**a**) Comparison of public key size. (**b**) Comparison of signing key and tracking key size.

To further demonstrate the advantages of the GS-MR scheme, we performed a progressive efficiency analysis and verification parameter size comparison between three lattice-based group signature schemes and the GS-MR scheme. They were, respectively, the group signature scheme with indexed attribute-based signature (ABS) proposed by Katsumata et al. [24], the group signature scheme with forward security and constant size proposed by Canard et al. [27], and the lattice-based dynamic group signature scheme proposed by Huang et al. [19]. In Table 4, we compare these three group signature schemes [19,24,27] with the GS-MR scheme, where $\lambda$ is the security parameter and $N = 2^l = poly(n)$ is the number of group members. From Table 4, it can be seen that the group public keys

in [19,24,27] are all related to the maximum number of group members *N*, whereas the public and private keys in our scheme and [19] are fixed values. However, in this paper, we needed to use the lattice-based delegation algorithm to generate member signing keys, causing the signature length to be linearly related to *N*. However, none of the above compared schemes have message recovery in the verification phase.

**Table 3.** Signature and message–signature pairs size (in KB).

| Parameters | *N* = 128 | *N* = 256 | *N* = 512 | *N* = 1024 |
|---|---|---|---|---|
| Signature | 155 | 309 | 617 | 1233 |
| Message-Signature | 219 | 373 | 681 | 1296 |
| Signature | 229 | 457 | 913 | 1825 |
| Message-Signature | 293 | 521 | 977 | 1889 |
| Signature | 365 | 729 | 1457 | 2913 |
| Message-Signature | 429 | 794 | 1521 | 2977 |

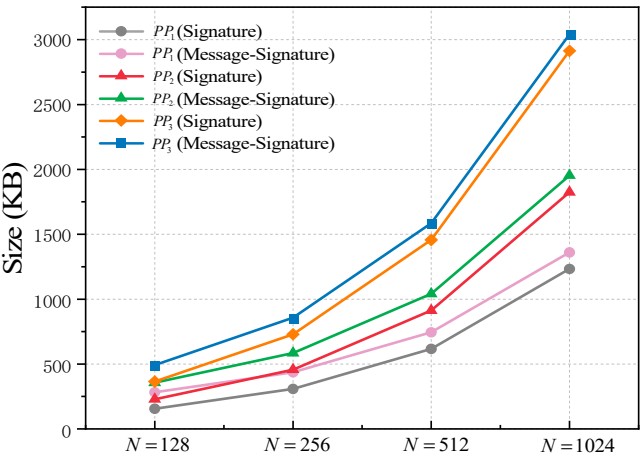

**Figure 4.** Comparison of the size of signature and message–signature pairs.

**Table 4.** Comparison of the progressive efficiency of group signatures.

| Scheme | Public Key Size | Private Key Size | Signature Size | Message Recovery |
|---|---|---|---|---|
| Katsumata [24] | $O(\lambda \cdot N)$ | $O(\lambda)$ | $O(\lambda \cdot N)$ | No |
| Canard [27] | $O(\lambda \cdot \log N)$ | $O(\lambda)$ | $O(\lambda)$ | No |
| Huang [19] | $O(\lambda^2)$ | $O(\lambda)$ | $O(\lambda \cdot \log N)$ | No |
| Ours | $O(\lambda)$ | $O(\lambda)$ | $O(\lambda \cdot N)$ | Yes |

For a more intuitive comparison, we chose fixed values for these schemes to compare the size of the verification parameter for different schemes under the same number of group members. We selected some fixed parameters while ensuring the security of the above comparison schemes. Let $n = 2^9, m = 4, q = 2^{27}, \sigma_1 \approx 7602.8121$, and $\sigma_2 = 1504.2467$. Finally, all schemes select a fixed message $M = \{0,1\}^{l_2}$ in the signature generation stage, where $l_2 = 2^{19}$. The comparison of the size of the verification parameters is shown in Table 5 and Figure 5.

**Table 5.** Comparison of the size of verification parameter.

| Scheme | Size of Verification Parameter (KB) | | | | |
|---|---|---|---|---|---|
| | *N* = 128 | *N* = 256 | *N* = 512 | *N* = 1024 | *N* = 2048 |
| Katsumata [24] | 741.80 | 1477.75 | 2949.75 | 5893.75 | 11781.70 |
| Canard [27] | 3033.75 | 3033.75 | 3033.75 | 3033.75 | 3033.75 |
| Huang [19] | 693.10 | 1033.10 | 1513.70 | 2673.70 | 3373.70 |
| Ours | 293.75 | 521.75 | 977.75 | 1889.75 | 3649.75 |

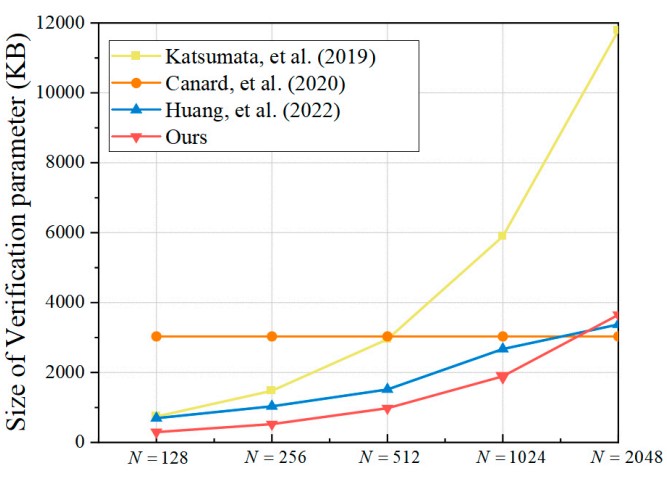

Number of group signers

**Figure 5.** The GS-MR scheme verification parameter size. Katsumata [24], Canard [27], Huang [19].

From Figure 5, it can be seen that compared to the other three schemes, the size of the verification parameters in the GS-MR scheme is the shortest when the number of group members $N \leq 1024$ is considered, with an average reduction of 53.02%. However, the signature size of GS-MR increases linearly with $N$, and at $N \geq 2048$, the verification parameter size is no longer advantageous compared to the Canard scheme [27] and the Huang scheme [27]. Overall, the proposed GS-MR scheme reduces the verification parameter size by an average of 39.17% compared to the schemes of [19,24,27].

## 7. Conclusions

In this paper, we proposed a lattice-based group signature with message recovery. The scheme achieves message recoverability in the validation phase, thus eliminating the need for group members to send additional messages as validation. And the GS-MR scheme ensures privacy and integrity in collaborative settings, benefiting applications where data security is crucial. It has potential applications in secure data sharing, blockchain systems, and federal learning. Then, we prove that the GS-MR scheme achieves full anonymity and traceability properties based on the difficulty of RSIS and RLWE problems. We also performed some experiments to evaluate the sizes of key and signature. Finally, we compare the GS-MR scheme with three group signature schemes and the result shows that the verification parameter of the GS-MR scheme was reduced by an average of 39.17% for less than 2000 members. Constructing a group signature scheme with controlled linkability under the quantum oracle model will be an attractive research topic for the future.

**Author Contributions:** Conceptualization, Y.T. and D.P.; methodology, D.P.; validation, D.P. and Y.T.; formal analysis, D.P. and L.L.; writing—original draft preparation, D.P.; writing—review and editing, Y.T. and P.Q.; supervision, Y.T.; funding acquisition, Y.T. and P.Q. All authors have read and agreed to the published version of the manuscript.

**Funding:** This research was partially supported by the Support Plan of Scientific and Technological Innovation Team in Universities of Henan Province (20IRTSTHN013) and the Henan Province Key R&D and Promotion Special Project (No.212102210166).

**Institutional Review Board Statement:** Not applicable.

**Informed Consent Statement:** Not applicable.

**Data Availability Statement:** Not applicable.

**Conflicts of Interest:** The authors declare no conflict of interest.

### Glossary: Symbol Definitions

| Notations | Explanation |
| --- | --- |
| $\mathcal{R}_q$ | Polynomial ring $\mathcal{R}_q = \mathbb{Z}_q[x]/(x^n+1)$ |
| $\tilde{A}$ | Gram-Schmidt orthogonalization of matrix $A$ |
| $\|x\|$ | Bit length when identifying $x$ with binary |
| $\|x\|^n$ | Takes $n$ bits from the high binary bit $x$ to the low bit |
| $\|x\|_n$ | Takes $n$ bits from the low binary bit $x$ to the high bit |
| $a \otimes b$ | The convolutional computing of two polynomials: $(a \cdot b)/<x^n+1>$ |
| $rot(a)$ | Circular matrix of $a \in \mathcal{R}_q$ |
| $\tau(a)$ | Vector of coefficients of the polynomial $a \in \mathcal{R}_q$ |
| $\tau^{-1}(a)$ | Transformation of the vector $a \in \mathbb{Z}^n$ into the corresponding polynomial |

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
