# Peer review of "Lattice-Based Group Signature with Message Recovery for Federal Learning"

_applsci, doi:10.3390/app13159007_

Round 1

Reviewer 1 Report

1.      Suggest adding a unified label to the formulas presented separately in the paper to facilitate readers' reference.

2.      The figures and tables presented in the paper should be labeled with the scheme name or abbreviation to help readers better identify them.

3.      The GS-MR scheme proposed in this article meets the requirements of complete anonymity and complete traceability. In section 3.2 of the security model, two games are described to characterize the security characteristics of the scheme. However, the specific process of the two games has been described in the security analysis, so I suggest simplifying the description of the two-game models and drawing them in the form of graphs for easy viewing.

Reviewer 2 Report

Novelty of the paper is low. The subject of this paper is not interesting. Quality of its presentation must be improved. New related references must be added. Therefore, I can not recommend it to publish.

Moderate editing of English language required

Reviewer 3 Report

The authors present a GS-MR-based signature anonymization algorithm and provide test results. Considering the importance of privacy, this study is quite interesting. The results obtained are impressive reduction. The authors can cite more relevant references in the literature review and elaborate on the problem more clearly for easy understanding. Section 3.2 has some gaps and they can be addressed by a minor revision. 

The authors have presented the paper interestingly and no english correction is required in my opinion.

Reviewer 4 Report

The article is clear, the literature references are sufficient, and the results are supported by examples. Experimental results are presented to highlight and validate the proposed approach with the support of two case studies. But the literature review and background are not sufficient, 

Reviewer 5 Report

This paper emphasizes the inseparable relationship between federal learning and privacy protection, highlighting the need to safeguard the privacy of participants in federated learning while also acknowledging the potential for privacy attacks using this technique. To address these concerns, the study proposes a lattice-based group signature with message recovery (GS-MR) as an effective tool for preserving user privacy in federated learning. The proposed GS-MR scheme is proven to offer full anonymity and traceability under the random oracle model, with anonymity and traceability reduced to the hardness assumptions of ring learning with errors (RLWE) and ring short integer solution (RSIS) respectively. The authors claim a significant reduction in message-signature size.

The work is presented in the security and privacy domain using machine learning and demonstrates a well-structured methodology. The article holds promise for a wide range of readers who are interested in this field. However, certain concerns should be addressed to enhance the overall quality of the research.

1.     In the Introduction, providing more contextual background information on federated learning, including its key principles, applications, and potential privacy concerns would be helpful. This would provide a stronger foundation for readers who may not be familiar with the topic. Moreover, it would be beneficial to explicitly state the research gap or problem statement that the proposed GS-MR scheme aims to address. This would help set the stage for understanding the novelty and significance of the research.

2.     The manuscript's structure description is missing at the end of the introduction. Please add a paragraph mentioning the organization of your manuscript.

3.     A brief literature review of related work in the field of lattice-based group signatures and message recovery schemes could be included to demonstrate the existing knowledge and approaches in the area, emphasizing the uniqueness and contribution of the proposed GS-MR scheme.

4.     Describe Table 5 and Figure 2 (b) in the text.

5.     The conclusion could reiterate the main contributions of the research in a concise and impactful manner. Elaborate on the practical implications and potential applications of the GS-MR scheme in real-world scenarios, particularly in federated learning or other contexts where privacy and integrity are crucial. This will help readers understand the practical value and relevance of the research. Provide a brief discussion on possible future research directions

6.     Proofread the entire manuscript to remove typos and grammatical errors.

7.     Include a few latest articles in the area of relevance.

Overall the quality of English is good.

Round 2

Reviewer 2 Report

This article has been significantly changed and is acceptable with minor changes

To make the article more comprehensive, it may be beneficial to discuss other encoding methods (hardware encoding methods) in the introduction. The following articles are recommended for reference.

1.          Hayati, M, Majidifar, S, Sobhani, SN. Using a hybrid encoding method based on the hexagonal resonators to increase the coding capacity of chipless RFID tags. Int J RF Microw Comput Aided Eng. 2022; 32 (12): e23474. doi:10.1002/mmce.23474.

2.          G. Karimi, S. Majidifar, A novel chipless RFID tag using spiral resonator to achieve the pentamerous data encoding form, J. Electromagn. Waves Appl. 28 (2014) 13–27. https://doi.org/10.1080/09205071.2013.854178.

Good
